# Towards a Framework for Better Understanding of Quiescent Cancer Cells

**DOI:** 10.3390/cells10030562

**Published:** 2021-03-05

**Authors:** Wan Najbah Nik Nabil, Zhichao Xi, Zejia Song, Lei Jin, Xu Dong Zhang, Hua Zhou, Paul De Souza, Qihan Dong, Hongxi Xu

**Affiliations:** 1School of Pharmacy, Shanghai University of Traditional Chinese Medicine, Shanghai 201203, China; najbah@yahoo.com (W.N.N.N.); xizhichaohaerbin@163.com (Z.X.); zj.suvisong@outlook.com (Z.S.); 2Pharmaceutical Services Programme, Ministry of Health, Petaling Jaya 46200, Malaysia; 3School of Medicine and Public Health, The University of Newcastle, Newcastle, NSW 2308, Australia; lei.jin@newcastle.edu.au; 4School of Biomedical Sciences and Pharmacy, The University of Newcastle, Newcastle, NSW 2308, Australia; xu.zhang@newcastle.edu.au; 5Shuguang Hospital, Shanghai University of Traditional Chinese Medicine, Shanghai 201203, China; zhouhuam@hotmail.com; 6School of Medicine, Western Sydney University, Sydney, NSW 2751, Australia; P.DeSouza@westernsydney.edu.au; 7Chinese Medicine Anti-Cancer Evaluation Program, Greg Brown Laboratory, Central Clinical School and Charles Perkins Centre, The University of Sydney, Sydney, NSW 2006, Australia; 8Department of Endocrinology, Royal Prince Alfred Hospital, Sydney, NSW 2050, Australia

**Keywords:** quiescence, dormancy, cancer, model, detection

## Abstract

Quiescent cancer cells (QCCs) are cancer cells that are reversibly suspended in G_0_ phase with the ability to re-enter the cell cycle and initiate tumor growth, and, ultimately, cancer recurrence and metastasis. QCCs are also therapeutically challenging due to their resistance to most conventional cancer treatments that selectively act on proliferating cells. Considering the significant impact of QCCs on cancer progression and treatment, better understanding of appropriate experimental models, and the evaluation of QCCs are key questions in the field that have direct influence on potential pharmacological interventions. Here, this review focuses on existing and emerging preclinical models and detection methods for QCCs and discusses their respective features and scope for application. By providing a framework for selecting appropriate experimental models and investigative methods, the identification of the key players that regulate the survival and activation of QCCs and the development of more effective QCC-targeting therapeutic agents may mitigate the consequences of QCCs.

## 1. Introduction

Despite recent advances in cancer treatment, patients still succumb to recurrence and metastasis, which are the main causes of cancer-related mortality [1,2]. Quiescent cancer cells (QCCs) transiently exit the cell cycle to reside in G_0_ phase [3,4] and, depending on the triggers that promote re-entry into the cell cycle, their behavior can affect both the clinical course of the disease and treatment effectiveness. The G_0_-G_1_ reversibility of QCCs distinguish them from other nonproliferating cells, such as senescent cells that are irreversibly arrested [5]. Cells enter quiescence state to survive deficiency of nutrition and growth factor, while continuous stimuli, including activated oncogene, impaired mitochondrial function, and DNA damaging agents induce the affected cells to senescence, leading to subsequent cell death [5,6]. Quiescent cells on the contrary, can re-enter the cell cycle and re-proliferate, thus evading cell death [6].

The presence of QCCs is common, and evident in various tumor types [1,4,7,8], suggesting that they are a fundamental property of cancer independent of histology. As the tumor develops in patients, cancer cells grow uncontrollably as they lose the “contact inhibition” property. The feature of “contact inhibition” in normal cells restrict the growth of a cell upon contact [9]. Cancer cells grow continuously even outreached the blood vessels and they become nutrient- and oxygen-deprived as they reside further from blood vessels. Entering into quiescence state allow these cancer cells to survive the oppressive environment [10]. The experimental QCCs models, such as nutrient deprivation, hypoxia-induced, and contact inhibition model, recapitulate the nutrient-, oxygen-deprived, and contact inhibition aspect of QCCs, respectively.

QCCs are similar to quiescent CSCs, in terms that they are both cancer cells in G_0_ phase. However, QCCs are distinct than cancer stem cells (CSCs) as CSCs can exist in any phase in the cell cycle, not necessarily in G_0_ phase. While quiescent CSCs are located at endosteal bone surface, QCCs are evident in tumor masses and everywhere as circulating tumor cells (CTCs) or disseminating tumor cells (DTCs) are frequently in quiescence. Moreover, self-renewal, “stem-ness” markers and specific transcription factors, which are expressed by the CSCs, but are absent in QCCs [1]. In addition to being in G_0_ phase and nonproliferating, other properties of QCCs are having less RNA content [11,12] and expressing Ki-67 negativity [13]. These characteristics of QCCs are the basis of the markers used to detect QCCs, for example Ki-67 that are applied in preclinical studies [14,15,16].

QCCs are nonproliferating, thus marking them as resistant to most conventional cancer treatments that act preferentially on proliferating cells [4]. Surviving QCCs can re-enter the cell cycle when conditions are suitable [3], and reproliferation gives rise to cancer progression and recurrence [17]. In addition, it is well-known that dissemination can occur early in the malignant process, but the basis for dormancy at secondary sites is cellular quiescence [18]. When QCCs survive in the local niche and are reactivated, clinically detectable metastases become apparent [1,7]. 

Therapeutic strategies targeting QCCs include blocking QCCs from re-entering the cell cycle, encouraging timing of therapies dependent on cell proliferation to match cell cycle re-entry points, or eradication of QCCs while in the G_0_ state. A rational understanding of these approaches, therefore, requires relevant models recapitulating QCCs behavior, as well as superior methods to evaluate QCCs activity. This review emphasizes the existing and emerging models of studying and measuring QCCs and discusses their respective features and applications.

## 2. Models Mimicking Quiescent Cancer Cells

Experimental models can be established through the use of either a homogenous or a heterogeneous QCC environment (Figure 1). Homogenously altered conditions of cell culture, including growing cancer cells until contact inhibition, depriving cell cultures of serum, nutrients or oxygen, induce the cultured cells to attain quiescence, thus enriching the culture with QCC. On the other hand, coculturing different cell populations (malignant and non-malignant) to mimic the tissue microenvironment results in models heterogeneously composed of proliferating and quiescent cells.

### 2.1. Homogenous In Vitro Models of QCC

#### 2.1.1. Serum or Glucose Deprivation

Prior to the “restriction point” in early G_1_ phase, mitogens, such as insulin-like growth factor-1 (IGF-1) [19] and platelet-derived growth factor (PDGF) [20], are removed in serum-starved cells and cause cells to exit the cell cycle [3]. Serum withdrawal of 7 days yielded 71% LNCaP prostate cancer cells in G_0_ phase, compared to 15% of cells cultured with full medium. Serum replenishment for 24 h, after serum withdrawal, released 50% of cells from the G_0_ phase [12]. A serum deprivation model has been performed on an extensive range of human and animal cell lines, as described in Table 1 [11,12,19,21,22,23,24].

The experimental protocol of serum deprivation varies considerably, suggesting that cross-study comparisons may not be valid. Cells are generally serum-deprived for 42 h [21] to 7 days [12,16,22], with a longer period of serum deprivation period correlating with more QCCs [16], and subsequently, more time required for QCCs to re-enter the cell cycle [20]. Although most studies involving serum deprivation use the serum-free medium method [11,12,16,19,22,25], low serum media containing 0.05% [21]—0.2% [21,23] is also employed. 

Serum deprivation-induced quiescence involves changes in multiple signaling molecules, including pathways involving retinoblastoma protein (Rb) [11,12,19], c-Myc [12], Mirk [11,19], p27 [11,19], and cytosolic phospholipase A_2_α (cPLA_2_α) [16]. Many of these proteins have been established as markers for quiescent cells, and these changes are also observable in other non-serum deprivation-induced QCC models [19].

To sustain cellular energy, glucose is required by proliferating cancer cells [19]. Therefore, a low glucose environment induces cancer cells to quiescence experimentally. Culturing TOV21 and SKOV3 ovarian cancer cells for 3 days in a low-glucose medium yielded approximately 80% and 65% G_0_ cells, respectively [19].

#### 2.1.2. Hypoxia-Induced Cell Quiescence

When the oxygen supply is insufficient, tumor cells survive by entering the G_0_ phase and may return to the G_1_ phase when oxygen is replenished [10,15]. Cancer cell lines can be experimentally exposed to hypoxia [8,15,27,28] through the use of a hypoxia chamber [8,15,26,27,28] or treatment with CoCl_2_ (concentration ranges from 100 μM to 500 μM) [15]. The degree of hypoxia in hypoxia chamber is severe, typically 1% oxygen [26,27,28] or 0.1% oxygen [15] for 5 [26] to 14 days [27]. The disadvantage of the hypoxia chamber approach is that cells re-establish normoxic behavior when removed from the chamber for subsequent assays or drug administration [15]. Hypoxic conditions are also mimicked by CoCl_2_ through induction of hypoxia-inducible factor-1 (HIF-1). When CoCl_2_ solution is withdrawn from the media, the cells reproliferate more slowly than in a hypoxia chamber, as additional time is required to remove the remaining CoCl_2_ from the cells. Compared to hypoxia chambers, the CoCl_2_ method offers a more sustainable oxygen-deficient environment and easier administration in in vivo animal models [15].

The hypoxia model is useful to mimic cancer interventions that act through hypoxia or ischemic mechanisms [28]. For example, trans-arterial chemoembolization (TACE), the standard of care for hepatocellular carcinoma, blocks the oxygen supply to hepatocellular tumor cells and triggers tumor cells to enter quiescence [28]. 

#### 2.1.3. Contact Inhibition at High Cell Density

A quiescent state can also be induced in cultured cells through contact inhibition. When a cell culture [12,22,24,30] reaches confluence, the contact between the cells can inhibit proliferation. Redistribution of these contact-inhibited cells at lower density allows them to re-cycle [29], which is the well-known basis for ‘splitting’ cell cultures to maintain live cell cultures in a laboratory. However, this model is unsuitable for nonmonolayer cell lines, such as LNCaP prostate or MIA PaCa-2 pancreatic cancer cells, or cells that continue to proliferate even in confluent states, such as HeLa cervical [24], A172 glioblastoma [30], or U251 glioma [30] cells.

#### 2.1.4. Anticancer Treatment Increase Proportion of QCC

The proportion of QCCs increase when cancer cells are exposed to cisplatin [17], doxorubicin [17,40,52], paclitaxel [17,37], 5-fluouracil [38], gemcitabine [23], or radiation [52]. The cancer treatment raises the proportion of QCC plausibly through killing the majority of proliferating cancer cells and/or inducing the proliferating cancer cells to quiescence. The underlying mechanism of QCC induction varies, depending on the treatment: doxorubicin induces QCCs via autophagy [40], while 5-fluouracil leads to QCCs by triggering c-Yes tyrosine kinase [39]. 

This model closely resembles QCC induction following cancer treatment in an actual clinical setting and is, thus, ideal for screening or testing potential drugs to reduce the high proportion of QCCs following anticancer treatment. However, the number of surviving QCCs in this model can be inadequate for further studies. For example, doxorubicin-treated mouse mammary carcinoma (MMC) cells underwent apoptosis, and only 31% of MMC cells remained viable 3 weeks posttreatment. Of these viable cells, 22% were QCCs compared to 3% QCCs in control MMC cells [52].

### 2.2. Heterogenous In Vitro Models of QCC

#### 2.2.1. Inner Layer of Spheres in Three-Dimensional (3D) Culture

On the evidence noted above, the proximity of cancer cells to blood vessels would be expected to contribute to tumor heterogeneity. Quiescence is readily promoted in cells furthest from blood vessels since there is less access to nutrients and oxygen and richer production of acidic metabolites [10]. Similarly, 3D sphere cultures mimic tumor heterogeneity, as the outer layer of 3D spheres has numerous proliferating cells, while the inner layer has QCCs, and the sphere core is mainly comprised of necrotic cells [10,35]. When the tumor spheres are exposed to conventional therapies that largely act on proliferating cells, the outer layer becomes nonviable, while the inner QCC layer resists conventional therapies [10]. Hence, this model is suitable to screen for potential QCC cytotoxic agents, such as compound VLX600, which can eradicate QCCs of colon cancer [53].

A 3D cell model can be constructed through scaffold or non-scaffold techniques. In the scaffold method, cancer cells are seeded on a matrix, such as basement membrane extract (BME) [32], poly (2-hydroxyethyl methacrylate) matrix [26,34], and agarose [33]. These matrices provide mechanical support that allows cancer cells to grow naturally as spheroids [35]. In the non-scaffold method, cancer cells are seeded in suspension, where 3D cell spheroids emerge via agitation, force-floating or the hanging-drops [35]. 

Three-dimensional models yield irregularly sized spheroids, even within the same well or flask [10,35], which may influence the response in drug screening [10]. Spheres < 400 μm in diameter can be utilized for cell viability assays because size is closely related to the number of viable cells [54]. Larger spheres ranging from 700 μm [31]—1200 μm [55] in diameter may be used to study histopathology [55] or to resemble the tumor phenotype at advanced stages [36]. However, studies on QCCs in 3D culture as a response to anticancer treatment are still lacking.

#### 2.2.2. Coculture Models

One of the drawbacks of conventional cell culture models is that they are unable to adequately mimic the microenvironment of actual tumor cells, thus restricting exploration of the effect of the microvascular niche on QCCs. To address this barrier, Ghajar and colleagues employed an organotypic coculture model by culturing human umbilical vein endothelial cells (HUVECs) in serum- and cytokine-free medium (SFM) to establish stable three-dimensional (3D) microvascular networks in SFM. HUVECs were then cocultured with fibroblasts from the lung or with bone marrow mesenchymal stem cells to construct a lung- or bone marrow-like microvascular niche. Thereafter, breast T4-2, MCF-7, and MDA-MB-231 cancer cells were seeded in these niches, and their growth was compared [42]. Mature endothelial cells express thrombospondin-1 (TSP-1), which then maintains quiescence of disseminating tumor cells (DTCs) in the lung or bone marrow. In contrast, the absence of TSP-1 in growing neovasculature hastened the outgrowth of DTCs [42].

Besides the organotypic coculture method described above, simple coculture models have been applied to examine the impact of the tumor environment niche on the quiescence or reactivation of prostate cancer cells [43] and breast cancer cells [44,45]. The partner cocultured cells vary; a hematopoietic stem cell niche [43], and the application of microRNAs and exosomes that are released from bone marrow stroma have been employed [44,45]. Compared to a prostate cancer PC-3 cell culture alone, the binding of PC-3 cells to the cocultured murine bone marrow-derived stromal cell line ST2 shielded PC-3 cells from chemotherapy-induced cell death and triggered mRNA expression of TANK (TRAF-associated NF-κβ activator) binding kinase 1 (TBK1), and PC-3 cells eventually became quiescent [43]. Coculturing breast MDA-MB-231 and T47D cancer cells with stroma expanded the pool of cancer cells residing in G_0_ phase, primarily attributable to passaging of relevant miRNAs through gap junctional intercellular communication (GJIC) between breast cancer cells and bone marrow stroma. Additionally, the quiescence of breast cancer cells is secondarily due to the transfer of exosomes containing miRNAs from bone marrow stroma to breast cancer cells [44]. Scrutinizing the role of the microenvironment on the process of quiescence and reactivation of cancer cells is, therefore, feasible by utilizing this coculture model.

### 2.3. In Vivo Quiescent Models

We herein describe some in vivo models that can recapitulate the interactions between cancer cells and the systemic host response during the quiescence process (Figure 2). To examine the dissemination of dormant cancer cells to metastatic sites, such as the lung, Barkan and colleagues injected breast MCF-7, MDA-MB-231, D2.0R, and D2A1 cancer cells intravenously into BALB/c nude mice [48]. D2A1 and MDA-MB-231 cells developed lung metastases within 2 to 3 weeks, whereas D2.0R and MCF-7 cells remained dormant in the lung until metastases were detected at 12 weeks and 6 months post-injection, respectively. Hence, intravenous injection of D2.0R and MCF-7 cells was proposed as an in vivo model of dormant lung metastases of breast cancer [48]. Their findings suggest that autophagy assists in the survival of quiescent breast cancer cells and that repressing autophagic flux causes reactive oxygen species (ROS) to accumulate, leading to cell apoptosis [32]. Using the same model, Barkan et al. later showed that fibrosis triggered dormant D2.0R cells to proliferate and develop metastases [51], and Weidenfled et al. reported that the expression of lysyl oxidase-like 2 (LOXL2) in quiescent MCF-7 cells allowed them to exit quiescence and cause metastases [56].

In investigating the impact of the environmental niche on quiescent cells, immunocompetent BKAL mice [47] and immunocompromised NOD/SCID (non-obese diabetes/severe combined immunodeficiency) mice [50] were either intravenously injected with labeled cancer cells or subcutaneously implanted with 3D-biomatrix [47,50]. The NOD/SCID mouse study on quiescent breast cancer cells involved coculturing the breast cancer cell line MDA-MB-231 with either a proliferation-promoting niche or a proliferation-inhibiting niche. Primary human bone marrow stromal cells in the 3D biomatrix function as a proliferation-promoting niche, while a biomatrix that consisted of a mixture of mesenchymal cells, osteoblast cells and endothelial cells served as a proliferation-inhibiting niche. Tumorigenesis was only observed on the supportive niche 8 weeks post-inoculation, but not on the inhibitory niche. Cancer cell colonies from the inhibitory niche were confirmed to be dormant and could reproliferate under suitable conditions. This model illustrates the influence of environmental niches on the quiescence of breast cancer cells [50]. Another in vivo example in examining factors involved in modulating quiescence of cancer cells at distant site is demonstrated by Yumoto and colleagues. To determine that Axl tyrosine kinase receptor is required to induce prostate cancer cells quiescent in bone marrow, SCID mice were intratibially injected with luciferase-labeled prostate cancer cell line (PC-3, DU145) that were either Axl-knockdown (sh Axl) or not (sh control). After tumor lesions were evident in the mice, examination of bone sections showed that knockdown of Axl significantly lowered the amount of quiescent prostate cancer cells and amplified apoptotic prostate cancer cells in the bone marrow, compared to control mice. These findings indicate the role of Axl in mediating quiescence and survival of prostate cancer cells in bone marrow [46].

Tumor dormancy was also studied through orthotopic injection of breast MDA-MB-231 cancer cells into NOD-SCID mice [42]. The primary tumor was excised 3 weeks post-inoculation. At 9 weeks post-inoculation, dissection of surviving mice found that breast cancer cells had disseminated to the lung, bone, liver and brain. Additionally, breast cancer cells that lodged in the microvascular endothelium of the lung and bone marrow became quiescent [42]. Injecting weakly metastatic breast cancer T4-2 cells intracardially into NOD-SCID mice led to quiescent T4-2 cells settling in the perivascular space of the lung, bone marrow and brain 8 weeks post-injection [42]. Quiescent tumor cells were evident in both intravenous and orthotopic routes of tumor inoculation, and their residing locations inferred the function of the endothelium in mediating tumor dormancy. 

In addition to dormancy of solid tumor, in vivo models can be established for hematological malignancies. To induce dormancy in mouse B cell lymphoma model (BCL1), BALB/c mice were first intraperitoneally injected with spleen cells from BCL1 tumor-bearing mice, followed by injection of IgM on day 28, 38, and 45 post-inoculation. At day 60 post-inoculation, mice immunized with IgM had smaller tumor growth than mice non-immunized with IgM. This model shows that anti-idiotype antibody is produced as response to injected IgM, and this antibody involves in the signaling maintaining dormancy [49]. The biology of BCL1 lymphoma model was further discussed in another review [57].

The use of labels, such as green fluorescent protein (GFP) [46,48,50,51], enhanced GFP (eGFP) [47], luciferase [46], and Discosoma sp. red fluorescent protein (dsRED) [50], enable the study of whether seeded cells are still maintained within the studied structure [50], monitoring tumor growth [46], and determining whether metastatic lesions are of solitary tumor cells or multicellular [51]. Applying lipophilic membrane dyes, such as 1,1′-dioctadecyl-3,3,3′,3′-tetramethylindo-dicarbocyanine (DiD), allows for the detection of dormant breast, prostate, and multiple myeloma cancer cells in the bone niche, as well as the sorting of these cells from harvested organs/tissues [47,58]. The journey of dormant cells at single-cell resolution can be tracked with technologies, such as intravital two-photon microscopy [47], in addition to examining the interaction between the cancer cell and bone environment that has resulted in dormancy or tumor outgrowth [58]. To differentiate between QCCs and normal quiescent cells in tissue, the morphology of the cells and the expression of oncogenes are first examined to discern the cancer cells [43,59]. Then, QCCs markers, such as Ki-67, are applied to detect the QCCs amongst proliferating cancer cells [42]. Another application of an in vivo study is to examine the movement of QCCs residing in or retreating to a quiescence state by manipulating potential pathways or proteins [7].

For the typical 2D in vitro culture, the medium contains adequate nutrients, growth factor and oxygen, allowing cancer cells to proliferate faster. However, these elements are less available in in vivo, causing slower tumor growth and higher proportion of QCCs [35,60]. In vivo studies are essential, as various biological aspects of the role of QCCs in cancer progression can be investigated in detail [7]. However, a major problem in studying quiescence-related genomic changes is the detachment of QCCs from their niches, since that can trigger cell cycle entry within minutes [61].

### 2.4. Discussion

Quiescence is not a homogenous phase but instead represents a diversified state [3]. Various factors (e.g., serum deprivation, contact inhibition, loss of adhesion) promote quiescence with different, yet overlapping gene expression profiles. The dissimilar gene profiles potentially explain the diverse quiescence phase that cells enter following triggering signals [20]. The induction period also determines the depth of the quiescent state. Longer serum-deprived conditions (6 days) or contact-inhibited (20 days) fibroblasts progress into a deeper quiescent state than shorter induction periods (2 days of serum deprivation or 4 days of contact inhibition) [3,20]. Cells in a deeper quiescent state are less likely and slower to re-enter the cell cycle upon growth stimulation [3]. The shallow quiescent state has been described as a “quiescence alert, G_Alert_”, midway between deep quiescence and activation. G_Alert_ cells have higher transcriptional activity and mitochondrial metabolism, larger cell size, and faster cell cycle re-entry than deep quiescent cells [62].

Considering that different quiescence induction stimuli induce varying gene profiles [20], the model of choice to establish QCCs should reflect the studied QCC conditions. In screening or testing potential therapeutic agents for cancer treatment-induced QCCs, QCCs should be induced by administering the cancer treatment used in the actual clinical setting. Another consideration is whether the cell line of choice can withstand the quiescence induction stimulus. Generally, in harsh environment (such as serum deprivation), un-phosphorylated pRb [62] and increased levels of cyclin-dependent kinase (CDK) inhibitors [3] can induce cells to enter and maintain their quiescence state. However, cancer cell property, such as inherent low p130/Rb2 expression, in ovarian cancer cells OVCAR4 and OV90 cause the cells to undergo apoptosis, rather than becoming quiescent when the cells are serum-deprived [19]. Cervical cancer HeLa cells is another example that cannot withstand serum-deprivation and go apoptosis [24], whereas for hypoxia-induced QCC model, the selected cancer cell lines need to withstand hypoxic condition and subsequently induce to quiescence, such as the pancreatic AsPC1 cancer cells. Other pancreatic and colorectal cancer cell lines, such as DLD-1, COLO320, MIA PaCa-2, and PANC1, can only endure acute hypoxia for less than 7 days and experience cell death when treated for longer period [27]. Monolayer cell lines are suitable to induce QCCs through contact inhibition model, while nonmonolayer cell lines, such as LNCaP or MIA PaCa-2, or cells that continue to proliferate even in confluent states, such as HeLa [24], A172 [30], or U251 [30] cells, are not suitable for this model. Worthy of note, most experimental cancer cell lines are specially treated to become immortalized; thus, they may not be the best models to study QCCs in vitro and in vivo. This limitation is due to immortalized cancer cells may not recapitulate the actual behavior of QCCs, or because implanted exogenous-sourced cancer cells may not reflect the natural course of cancer in animal or human [63].

As actual solid tumors develop in vivo in a 3-dimensional manner, the 3D culture model may more closely resemble solid tumor biology, in terms of heterogeneous tumor composition [10], the microenvironment surrounding the tumor [10], cell-cell interaction [35], and cell-extracellular matrix interaction [35]. These similarities lead to comparable cellular responses [35] and gene or protein expression between 3D models and actual solid tumors [35]. Although the 3D model serves as a more reasonable model for clinical drug development than the 2D culture model, staining of the 3D model is challenging, as only small molecular weight dyes can be used for quantification and qualification purposes. Antibodies are bulky and difficult to use in 3D models [26]. Furthermore, methods to detect the response of quiescent cells in 3D models following interventions are lacking, and the response of QCCs is difficult to detect directly.

## 3. Measurement of Quiescent Cancer Cells

Currently, the identification of Ki-67 negativity and low cellular RNA content are commonly used to define QCCs. Other measurements, such as assessment of DNA content and potential molecular markers for quiescence, are often required as supplementary evidence to facilitate identification of cells at G_0_ phase from cells at other phases, especially G_1_ phase, because cells at G_0_ and G_1_ phases have the same DNA content [11] (Figure 3).

### 3.1. Ki-67

Ki-67, a nuclear protein present in proliferating cells, has been used as a clinical marker in cancer prognosis to reflect the proliferative index [13]. Ki-67 progressively degrades from M to G_1_ phase [13], resulting in Ki-67 negativity in quiescent cells [14]. Nevertheless, cells can still weakly express Ki-67 when exiting G_1_ to G_0_, and G_1_ cells surging from lengthy quiescence are still Ki-67 negative [13]. Of note, senescent and terminally differentiated cells are also Ki-67 negative or weakly express Ki-67 [13], and can be mistakenly identified as cells residing in the G_0_ state. Differed from quiescent cells, senescent cells can be identified based on its high senescence-associated β-galactosidase (SA-β-Gal) activity, presence of p16 and degradation of MDM2 [6]. To minimize false positive readings on G_0_ cells, the Ki-67 marker is regularly combined with other DNA stains (such as 4′,6′-diamino-2-phenylindole (DAPI) [15] and PI [14,15]) or cell cycle regulatory proteins, such as p27 [16,33] in vitro and in vivo studies.

### 3.2. RNA and DNA Content

Although quiescent and G_1_ cells have 2n DNA [62], quiescent cells transcribe less RNA content [11,12], and combinatorial measurement of RNA and DNA contents is instrumental in differentiating G_0_ from G_1_ phase cells [12]. Therefore, this approach may detect QCCs more accurately than single use of Ki-67.

Hoechst 33528, pyronin Y [11,12], acridine orange (AO) [26,29,64], 5-bromo-2′-deoxyuridine (BrdU), and 5-ethynyl-2′-deoxyuridine (EdU) [31,65] are commonly used to stain nucleic acids. Hoechst stains double-stranded DNA and distinguishes G_0/1_ cells from cells at other phases. Pyronin stains RNA and facilitates the identification of G_0_ cells that have lower RNA than G_1_ cells [11,12]. Cells can endure low concentrations of Hoechst 33528 and pyronin Y staining, allowing them to be sorted and cultured for subsequent experiments [66]. AO emits green fluorescence in double-stranded DNA and red fluorescence in single-stranded RNA, thereby distinguishing QCCs, which have a lower ratio of red/green fluorescence [26,64]. However, AO also stains lysosomes, emitting bright orange fluorescence [67]; thus, double staining Hoechst/Pyronin Y may be better in detecting QCCs. The DNA replication indicators BrdU and EdU are synthetic thymidine analogs that integrate into replicating DNA in cells at S phase [31,65], enabling tracing of the journey of dividing cells and their offspring [65]. Unlike BrdU, DNA denaturation is not required to detect EdU incorporation, thus enabling more sensitive detection, simpler protocols, and the feasibility of subsequent co-staining [14]. In contrast, BrdU is firmly incorporated into DNA and measurable even after months [68]. Propidium Iodide (PI) staining reflects DNA content and distinguishes diploid G_0/1_ cells from other S/G_2_/M phases. However, this necessitates the combined use of RNA staining or other markers to identify G_0_ cells [12,23]. 

However, the application of DNA and RNA content methodology with stains is more complex than the Ki-67 assay. An unsuitable concentration of AO affects the accuracy of determining the cellular DNA and RNA contents; excessive AO will denature DNA, while overdiluted AO will partially denature RNA [64]. Higher concentrations of Hoechst 33528 and pyronin Y are cytotoxic [66] and require strict control to avoid toxicity when used in vivo.

### 3.3. Fluorescence Ubiquitin Cell Cycle Indicator (FUCCI)

FUCCI utilizes the phase-dependent abundance of certain proteins, such as Cdt1 and geminin, to differentiate cells at different phases of the cell cycle. The fluorescent proteins mKO2 and mAG are fused to Cdt1 and geminin, respectively [69]. Cdt1 is present in G_1_ and G_0_ phase and gradually declines in subsequent phases, allowing mKO2-hCdt1 to mark G_0/1_ cells with red fluorescence. As geminin accumulates in S/G_2_/M phases, mAG-hGem labels S/G_2_/M cells green [69]. Cell phases can be distinguished by gating the FUCCI signals into 5 categories: mKO2^++^/mAG^-^, mKO2^+^/mAG^-^, mKO2^-^/mAG^-^, mKO2^+^/mAG^+^, and mKO2^-^/mAG^+^, which represent G_0_, G_1_, very early G_1_, G_1_/S, and S/G_2_/M phases, respectively [69]. Another FUCCI method applies to the combination of mCherry-hCdt1 with mVenus-p27K^-^, which is also instrumental in discerning G_0_ cells from G_1_ cells, as G_0_ cells express high levels of both mVenus-p27K^-^ and mCherry-hCdt1 [70].

### 3.4. Potential QCC Molecular Markers

Some proteins that are considerably up- or downregulated in QCCs have been considered QCC biomarkers. However, QCC molecular markers are nonexclusively present in G_0_ cells; therefore, the combination of these markers with the measurements mentioned in the previous section is advantageous in detecting QCCs. Increased hypoxic conditions within tumors has been well documented [10,15,28] and exploiting this is one means of inducing quiescence in in vitro and in vivo studies [8,15,27]. Hypoxic markers (such as pimonidazol [31,65] and HIF-1α [15,31]) mark hypoxic cells. However, to specifically detect hypoxic QCCs, QCCs markers (such as Ki-67) are applied together with hypoxic markers. Hypoxic markers can be applied in immunohistochemistry [27,31,65], immunofluorescence [15] or Western blotting analysis [15].

QCCs display p38^high^/ERK^low^ [15] and MCM2^low^/H3K9me2^low^/ HES1^high^ [37], high levels of p27 [11,19], DYRK1B [19] and NR2F1 [71], and low levels of c-Myc [12] and ROS [72]. In a study of QCCs induced by hypoxia, 2D and 3D breast cancer MCF-7 cells were treated with 100 μM CoCl_2_ for 72 h. Western blot analysis of both 2D and 3D cells recorded high p38 and p21, low ERK activity and low mRNA levels of c-Myc compared with untreated controls [15]. G_0_/G_1_ switch gene 2 (GOS2) is another potential QCC biomarker yet to be further explored, as GOS2 is highly expressed in G_0_ cells. GOS2 promotes quiescence in hematopoietic stem cells [73], increases the fraction of leukemia K562 cells in G_0_ phase and induces tumor regression in vivo [74].

### 3.5. Pulse-Chase Identification

Briefly, the pulse-chase method involves tagging fluorescent compounds onto cells in the pulse phase and chase phase, allowing the fluorescence to be diluted as a result of cell division. In the pulse phase, fluorescent markers, such as the proliferation marker carboxyfluorescein succinimidyl ester (CFSE) [75] or lipophilic dye Vybrant^®^ DiD [76], are tagged to colon HCT116 cancer cells and breast MCF-7, MDA-MB-231 cancer cells and then seeded [75,76]. The chase phase follows, whereby cells are cultured for 1 week [75] or undergo 6 consecutive passages [76]. Proliferating cells attenuate the fluorescence intensity in this chase phase [75,76]. Then, the cultured cells are subjected to subsequent live sorting for subpopulations of CFSE^+^ or DiD^+^ cells [75,76]. In contrast to diluted labeling dyes in proliferating cells, the dye-retaining features of slow cycling cells assist in demonstrating the enrichment of slow cycling cells posttreatment with chemotherapy regimens oxaliplatin and 5-fluorouracil [75]. In addition, membrane-labeling dyes, such as PKH26, PKH27, DiD, and DiI (1,1′-dioctadecyl-3,3,3′3′-tetramethylindocarbocyanine perchlorate), work on similar principle. After the membrane-labeling dyes diffuse to the cell membrane, their fluorescence intensity diminish as the cells divided because the dyes were distributed uniformly among the daughter cells [76,77,78]. The noncytotoxic nature of PKH dyes allow them to be used for examining live cells and functional assays on labeled cells for in vivo use [77]. Primary tumors and metastases have been confirmed to be heterogeneously composed of proliferating and slow cycling cells based on the PKH-67 label-retaining features in vivo [77]. To further identify quiescent cells among slow cycling cells, Hoechst/pyronin Y staining was applied [77].

With the pulse-chase method, live cells can be sorted for subsequent functional studies and comparisons with other cell subpopulations [76]. However, as the identified label-retaining cells are slow cycling cells that proliferate slowly due to S- or G_2_/M phase arrest [75,76], they do not necessarily reside at G_0_ phase, hence necessitating coupling this assay with G_0_ identification methods in identifying QCCs. An adequate culturing time period of up to several weeks is required to differentiate between non- and label-retaining cells [76]. Although pulse-chase analysis is feasible for adherent cultures, not all sphere cultures are suitable for this analysis, such as sphere cultures of MDA-MB-231 cells composed of laxly attached cell clusters [75].

### 3.6. Discussion

Better ways to characterize and detect QCCs are essential to bridge the gap between limited biological knowledge and clinical observation of QCCs. Current insights into QCCs have been largely drawn from static histology of static cytologic examination or retention of fluorescent dye, but a more advanced QCC detection method that allows examination of the dynamic process of QCCs is required. Quantum dot imaging, MRI labeling, and intravital microscopy are some examples of in vivo imaging methods that can be integrated into QCC studies to provide real-time observations on QCCs behavior [7]. Although each described QCC detection method has advantages and disadvantages, developing a superior ability to track QCCs may be assist in better delineation of tumor margins between benign and malignant tumors, and thereby guide treatment choice. For example, taking advantage of the fact that Ki-67 is not expressed in G_0_ phase, investigators were able to verify that Ki-67 levels were significantly lower in benign cells and nonatypical hyperplastic cells compared to atypical hyperplastic cells and malignant cells from endometrial samples [79]. Another promising clinical application of discerning quiescence is the potential prediction of metastasis-free periods in cancer patients [80].

In brief, Ki-67 alone or in combination with fluorescent dyes or other protein markers may detect and measure QCCs in preclinical studies. Ki-67 has been a reliable marker for QCCs [62], and determining the basal level of Ki-67 facilitates the study of the transition from G_1_, G_Alert_ and G_0_ phase, or vice versa. Other markers include the DNA intercalating agents BrdU and EdU, which are absent or poorly retained in quiescent cells. However, both BrdU and EdU are unsuitable for downstream and lengthy analysis because they harm cells and induce mutations [81]. Fluorescent dyes are categorized according to their binding affinity to: (a) nucleic acids, (b) cytoplasm, (c) membrane, and d) cell cycle-related proteins. Nucleic acid binding dyes, such as Hoechst 33528, pyronin Y, and AO, are commonly employed to analyze the cell cycle, cell migration and movement both in vitro and in vivo [82]. However, different RNA levels across tissues and cell types [61] cause disparities when distinguishing between G_0_ and G_1_ cells. Both nucleic acid-binding dyes and cytoplasmic dyes (e.g., CFSE) are not feasible for prolonged experiments because they disrupt cell proliferation [82] or are highly cytotoxic [82]. Conversely, membrane binding dyes (e.g., PKH26, PKH67, DiD, DiI) are less cytotoxic and their long retention in cells makes them useful in cell migration and proliferation studies [78,82], with DiD and DiI providing fewer confounding results than PKH dyes [76,78,82]. 

QCCs have been predominantly studied preclinically, and of the discussed QCC biomarkers, only Ki-67 has been clinically assessed and included in clinical practice oncology guidelines [2]. Contrary to preclinical studies that evaluate the fraction of quiescent cells [14,15,16], in the clinical setting, the Ki-67 index of biopsy samples predicts the proliferative rate of tumors, which serves as guidance in treatment selection, estimation of patients’ treatment response and clinical outcome [2]. Because QCCs can exist at primary and secondary tumor sites [4], even before metastases become clinically apparent [80], the development and refinement of QCC methodology that is safe and applicable to the clinical setting may facilitate the early detection, surveillance and treatment of cancer. The translation of QCC biomarkers identified from preclinical studies to human specimens will also allow for investigation of mechanisms promoting and inhibiting QCC that are relevant clinically. Tumors are heterogeneously composed of proliferating and quiescent cells [7], but biopsy samples assess only a fraction of tumor burden within an individual, and this selection bias may not accurately reflect tumor grading. Besides the discussed QCCs markers, single cell RNA-seq (scRNA-seq) is another potential technology applicable to reveal the heterogeneity present in QCCs and identify the QCCs more precise [83]. However, currently, limited research has explored the applications of scRNA-seq on identifying QCCs. For example, when Andor et al. examining the scRNA-seq of lymphoma patients, they presented the differed B cell scRNA-seq transcriptional profiles between proliferating and quiescent malignant B cells [84]. Another study conducted scRNA-seq to distinguish between quiescent and re-activated murine myeloma 5TGM1 cells that were previously injected into BKAL mice [47]. A combination of advanced imaging and highly sensitive and specific QCC measurements may aid in locating tumors or micrometastatic sites, determining tumor composition, assessing treatment response, and facilitating QCC-targeting clinical trials.

## 4. Conclusions and Future Perspectives

Cancer dormancy is a well-known clinical phenomenon that is attributable to QCCs. Reversibly residing in G_0_ phase [3] and being resistant to most conventional cancer treatments allows QCCs to survive [10]. When exiting the quiescent state and entering the cell cycle, activated QCCs result in cancer progression, recurrence [3,17], and clinically detectable metastases [18]. Despite its clinical significance, much about cell cycle re-entry remains undiscovered, and therapeutic options are rare. Breakthroughs in finding the key players regulating the survival and activation of QCCs, which are required for the development of QCC-targeting agents, will require relevant QCC models and quantification methods. The development of QCCs-targeting agents necessitates rational models that can recapitulate QCCs behavior in human to understand the role of QCCs in cancer progression and identify therapeutic opportunities. After QCCs models are established, detection method to trace QCCs activity is essential to characterize QCC, explore the underlying mechanisms, verify the hypothesized mechanism, and to test the response of developed QCCs-targeting agents.

We have provided a broad framework for assessing the relevance of QCC models, including (i) the suitability of the available models for investigated cancer types, (ii) the drawbacks, and (iii) possible opportunities for overcoming these obstacles. Because G_0_ cells are diploid, and it is not possible to separate them from G_1_ cells by DNA content, QCCs are detected and currently quantified through the absence of Ki-67 protein or low RNA content. Harnessing live cancer cells with a fluorescence-labeled G_0_ specific protein so that quiescence (gain of fluorescence) and cell cycle re-entry (loss of fluorescence) can be monitored under natural conditions without any experimental induction deserves serious consideration for future studies in this field.

## Figures and Tables

**Figure 1 cells-10-00562-f001:**
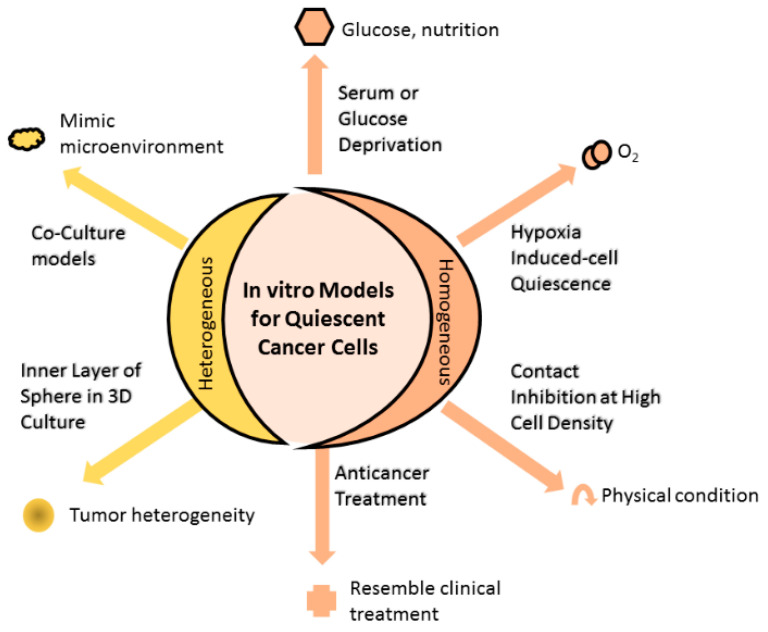
In vitro models for quiescent cancer cells (QCCs). Changing the physical condition of cell culture by allowing cultured cells to grow at high cell density until reaching contact inhibition or depriving essential substances (such as serum, glucose, or oxygen) of the culture, or administering anticancer treatment into cell culture that homogenously induces cells into quiescence. Alternatively, cell culture can heterogeneously comprise quiescent and proliferating cells using the 3D culture method. The inner and outer layers of 3D spheres are mainly composed of quiescent and proliferating cells, respectively, resembling the tumor heterogeneity observed in the clinical setting. Other means to achieve heterogeneous cell composition are coculturing different cell populations that mimic the microenvironment in which actual tumor cells grow, while anticancer treatment induces a portion of cells to quiescence, as found in the clinical setting.

**Figure 2 cells-10-00562-f002:**
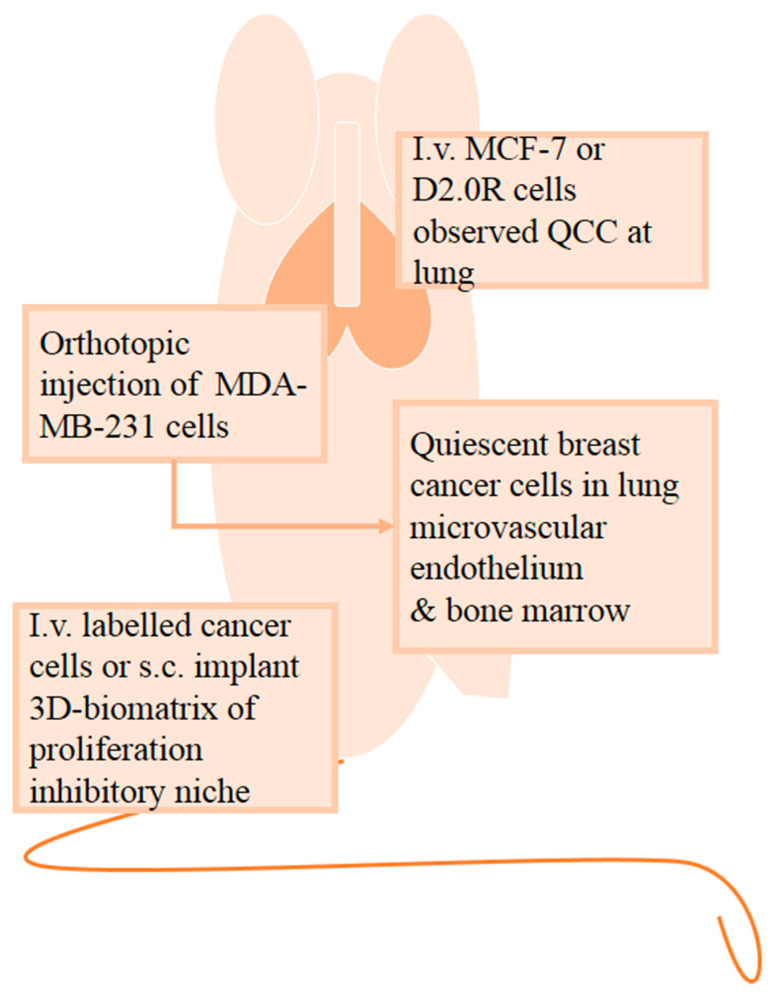
In vivo quiescent models are applied to examine the interactions between cancer cells and the systemic host response. To study the dissemination of quiescent cancer cells (QCCs) to metastatic sites, namely, the lung, mice were intravenously (i.v.) injected with breast cancer MCF-7 cells or mouse mammary carcinoma D2.0R cells. The effect of the environmental niche on QCCs can be examined by either intravenous injection of labeled cancer cells or subcutaneously (s.c.) implanting 3D biomatrix that serves as a proliferation inhibitory niche. Following orthotopic injection of MDA-MB-231 breast cancer cells into mice, QCCs were detected in the microvascular endothelium of the lung and bone marrow.

**Figure 3 cells-10-00562-f003:**
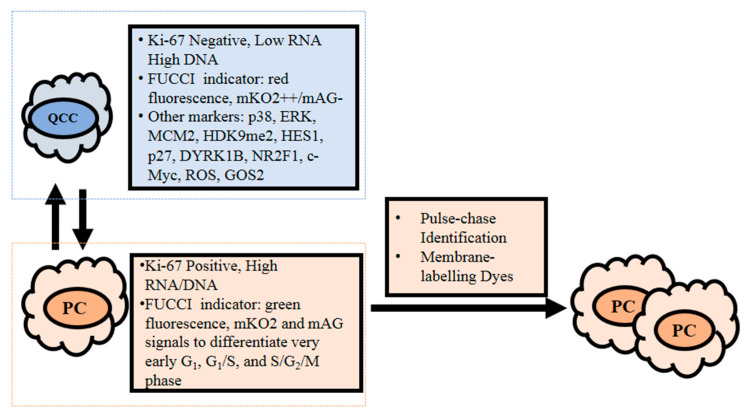
Current detection methodology for quiescent cancer cells (QCCs) that reside at G_0_ phase. QCCs are generally detected based on their Ki-67 negative and low cellular RNA content, while proliferating cells are Ki-67 positive and contain high RNA and DNA. Other detection methods are also applied to assist in differentiating between quiescent and proliferating cells. With a fluorescence ubiquitin cell cycle indicator (FUCCI), G_0_/G_1_ cells are labeled with red fluorescence or identified based on their mKO2^++^/mAG^-^ signal. Proliferating cells are identifiable by FUCCI as green fluorescence and further distinguished via mKO2 and mAG signals: G_1_ (mKO2^+^/mAG^-^), very early G_1_ (mKO2^-^/mAG^-^), G_1_/S (mKO2^+^/mAG^+^), and S/G_2_/M (mKO2^-^/mAG^+^) phases. Other potential molecular markers for QCCs are p38^high^/ERK^low^, MCM2^low^/H3K9me2^low^/HES1^high^; high levels of p27, dual specificity tyrosine phosphorylation-regulated kinase 1B (DYRK1B), nuclear receptor subfamily 2 group F member 1 (NR2F1), and G_0_/G_1_ switch gene 2 (GOS2); and low levels of c-Myc and reactive oxygen species (ROS). Pulse-chase identification and membrane-labeling dye methods involve dilution of respective fluorescent and membrane-labeling dyes as cells proliferate. Hence, label-retaining cells are slow cycling cells, while label-diluted cells are proliferating cells.

**Table 1 cells-10-00562-t001:** Overview of the model and method of quiescent cancer cell studies.

Model	Method	Cell Line	Detection	Molecules/Pathways	Remarks
Serum deprivation	Serum free [11,12,16,19,22] or low serum medium [20,21,23,24]	Prostate (LNCaP) [12,16,22], ovarian (SKOV3, TOV21G, OVCAR3) [19], lung (H1975) [22], pancreatic (SU86.86, Panc1, AsPc1) [23], colon (HD6) [11], renal (786-0) [24], multiple myeloma (NCI-H929, RPMI8226, U266) [21], osteosarcoma (MNNG-HOS) [25]	Ki-67 [16,22] Hoechst/Pyronin Y [11,12,19,21,22,23], PI [12,20,21,23,24], BrdU [16,20,24], p27 [16,19,24]	Rb [12,19], c-Myc [12], Cyclin D [19], Mirk [11,19], p27 [11,19], cPLA_2_α [16]	-Some cell lines are not suitable for induction to QCCs, e.g., OVCAR4, OV90, HeLa [19,24].
Nutrient deprivation	Low glucose [19]	Ovarian (SKOV3, TOV21G) [19]	Hoechst/Pyronin Y [19]	AMPK [19], cAMP [19], mTOR [19]	-With limited studies
Hypoxia	1% oxygen [15,26,27,28] or cobalt chloride [15]	Breast (MCF-7, MDA-MB-231) [15], pancreatic (AsPC-1) [27], ovarian (OVCAR-3) [15], liver (HepG2, SNU-449, SNU-387, SNU-398) [28] leukemia (KCL22, K562) [8], MEF [26]	HIF-α [8,15,27], Pimonidazole [27], Hoechst/Pyronin Y [8], Hoechst [28], Ki-67 [15,28], BrdU [27], PI [15,27], DAPI [15], p38 [27], ERK [27], acridine orange [26]	HIF [8,15], AKT [27], ERK [27] p21 [15], Myc [15], VHL [15], catenin [8]	-Tumor cell types have dissimilar resistance towards acute and chronic hypoxia [27]. -Suitable for cancer intervention that causes hypoxia [28].
Contact inhibition	Confluent culture cells [20,29]	Prostate (PC-3) [12,22], glioma (T98G, NAC6) [30], renal (786-0) [24], lung (A549) [22]	Ki-67 [22], Hoechst/Pyronin Y [12,22], PI [12,24,30], BrdU [24], mVenus-p27K [22], p27 [24,30]	C-Myc [12,22], VHL [29], pRb [12,22], p27 [22,30], FACT [22], Wnt [29], Cyclin D [29]	-Unsuitable for cell lines that continue to proliferate at confluent state [24,30]
3D culture	(a) scaffold method [31]–BME [32], agarose [33], poly (2-hydroxyethyl methacrylate matrix) [26,34] (b) non-scaffold method-agitation, force-floating or hanging-drop method [35]	Colon (DLD-1) [33], breast (MCF-7 and MDA-MB-231) [32]; tongue (SAS) [31], osteosarcoma (MHM, MNNG-HOS, SJSA-I) [34], mouse mammary cancer cell (D2.0R and D2A1) [32], MEF [26]	Ki-67 [32,33,34], FUCCI [31], HIF-α [31], Pimonidazole [31], Hoechst [31], EdU [31], DAPI [32], p27 [33], acridine orange [26]	Low oxygen and nutrient supply, accumulation of metabolite [10], HIF-1α [36], ATG7 [32], p27 [33], Wnt-β catenin [34]	-Irregular size of spheroids [10,35]. -Irregular proportion of QCCs in spheroids [35].
Cancer treatment	Chemotherapy or radiotherapy	Breast (MCF7, MDA-MB-231) [37], colon (HCT 116 [37,38], LoVo [38], HT29 [39]), lung (PC9) [37], melanoma (A375) [37], pancreas (SU86.86) [23], stomach (MKN45) [17]	Ki-67 [38,39], Hoechst/Pyronin Y [23], PI [23,38], DAPI [37], CFSE [38], FUCCI [17]	Autophagy [40], G6PD [41], Yes tyrosine kinase [39], Mirk [23]	-Resemble QCCs in tumors induced by treatments
Coculture	(a) organotypic coculture [42] (b) simple coculture [43,44,45,46,47]	Coculture endothelial cells (HUVECs) with breast cancer cells (T4-2, MCF-7, MDA-MB-231) [42]; coculture prostate PC-3 cancer cells with murine bone marrow stromal cell line ST2 cells [43]; coculture breast cancer cells (MDA-MB-231, T47D) with bone marrow stroma [44]; coculture prostate cancer cells (PC-3, DU145) with pre-osteoblastic cells (MC3T3-E1) [46]; coculture murine myeloma cancer cells (5TGM1) with pre-osteoblastic cells (MC3T3) [47]; coculture bone marrow-metastatic breast cancer cell (BM2) with bone marrow-mesenchymal stem cells (BM-MSCs) from human donor [45]	Ki-67 [42,43,46], Hoechst/Pyronin Y [44,46], DAPI [46], DiD [47], Hoechst [45], PKH26 [45]	TSP-1 [42], TBK1 [43], IKKE [43], miRNAs passage through GIJC [44], Axl [46,47], miR-23b [45]	-Mimic microenvironment in actual tumor, which allow for examining effect of microvascular niche [42] and tumor microenvironment [43] on QCCs
In vivo	Xenograft on BALB/c nude mice [48,49], BKAL mice [47], NOD/SCID mice [42,50], SCID mice [46]	Breast (MCF-7 [48], MDA-MB-231 [42,48,50], T4-2 [42]), mouse mammary cancer cell (D2.0R, D2A1) [48], prostate (PC-3, DU145) [46], murine myeloma (5TGM1) [47], lymphoma (BCL1) cell [49]	Ki-67 [42,46], DAPI [42,46,48], Hoechst [42], DiD [47], CFSE [49]	ATG7 [32], myosin light chain kinase (MLCK) [48], integrin β1 signaling [48], FAK [51], ERK [51], p38 [50], TGF-β [46,50], Axl [46,47]	-With limited studies

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
