# Peer review of "Towards a Framework for Better Understanding of Quiescent Cancer Cells"

_cells, 2021, doi:10.3390/cells10030562_

Round 1

Reviewer 1 Report

The review “Towards a framework for better understanding of quiescent cancer cells” presents the literature exploring quiescent cancer cells (QCCs) that are characterized by their G0-phase cell state, while they can still enter into cell cycle and initiate tumor recurrence and metastasis. This review is interesting because it is accompanied by a real reflection and is not a mere catalogue of literature; however some improvements are suggested.

Authors have chosen to expose in a first part the in vitro and in vivo models QCCs (“2. Models mimicking quiescent cancer cells”) and in a second part the methods used to identify QCCs (“3. Measurement of quiescent cancer cells”). Consequently they compared QCC identification in various models, but they did not mention the variability of detection methods used to identify QCCs in each report. It is indeed more interesting to start with biological results, stimulating the interest of the reader, before entering into a more technical and potentially boring part on the technical limits that can block the interest of the reader for biological results. To overcome this limitation without changing the overall organization of their review, authors may include a second table with the methods used to detect CQC for each reference presented in Part 1 and Table 1. Such a Table 2 would assist readers in selecting their reference material for their own benefit and would be a link between Part 1 and Part 2 of this manuscript.

Between lines 98 and 101, the authors grouped references 18 and 9, which have based their QCC identification on DNA and RNA staining, although using two different methods. The main critical point is that the authors did not mention that glucose deprivation had led to very different variations of QCC in both reports. No significant increase of quiescent cells has been observed following glucose depletion in the reference 18, whereas authors reported for the reference 9, that “low-glucose medium yielded approximately 80% and 65% G0 cells” in two ovarian cancer cell lines. Authors should improve their critical analyse on these two references presented in the same paragraph. They should not used them to suggest that both demonstrated that low glucose induces quiescent cells, while their results seem to be different.

Because the improvement of treatment in pediatric tumors is of high interest, I would suggest to add references for quiescent cancer cells in osteosarcoma, which is characterized by a strong resistance to treatment, recurrence and metastatic progression.

Author Response

Point 1: Authors have chosen to expose in a first part the in vitro and in vivo models QCCs (“2. Models mimicking quiescent cancer cells”) and in a second part the methods used to identify QCCs (“3. Measurement of quiescent cancer cells”). Consequently they compared QCC identification in various models, but they did not mention the variability of detection methods used to identify QCCs in each report. It is indeed more interesting to start with biological results, stimulating the interest of the reader, before entering into a more technical and potentially boring part on the technical limits that can block the interest of the reader for biological results. To overcome this limitation without changing the overall organization of their review, authors may include a second table with the methods used to detect QCC for each reference presented in Part 1 and Table 1. Such a Table 2 would assist readers in selecting their reference material for their own benefit and would be a link between Part 1 and Part 2 of this manuscript. 

Response 1: Thank you for pointing this out. To improve the link between QCCs models and detection method, we have inserted in Table 1 detection method used in each model.

Point 2: Between lines 98 and 101, the authors grouped references 18 and 9, which have based their QCC identification on DNA and RNA staining, although using two different methods. The main critical point is that the authors did not mention that glucose deprivation had led to very different variations of QCC in both reports. No significant increase of quiescent cells has been observed following glucose depletion in the reference 18, whereas authors reported for the reference 9, that “low-glucose medium yielded approximately 80% and 65% G0 cells” in two ovarian cancer cell lines. Authors should improve their critical analyse on these two references presented in the same paragraph. They should not used them to suggest that both demonstrated that low glucose induces quiescent cells, while their results seem to be different.

Response 2: We agree with the reviewer regarding reference 18 (Kim et al 2008) and have used reference 9 (Hu et al. 2011) to replace 18 in between lines 125 and 128.

Point 3: Because the improvement of treatment in pediatric tumors is of high interest, I would suggest to add references for quiescent cancer cells in osteosarcoma, which is characterized by a strong resistance to treatment, recurrence and metastatic progression.

Response 3: Thank you for the suggestion. As highlighted in Table 1, “2.1.1. Serum or glucose deprivation” (page 5, line 118) and “2.2.1. Inner layer of spheres in 3D culture” (page 6, line 185), we have added references numbered 25 and 34 about QCCs osteosarcoma.

Reviewer 2 Report

In their review paper ‘Towards a framework for better understanding of quiescent cancer cells’, Nabil, Xi et al discuss experimental models to study quiescent cancer cells (QCC) and the techniques used for QCC detection. The paper is quite clear and the information provided could help someone who is new to the field to quickly grasp the experimental in how to address QCC experimentally. Still, I would be nice if the authors could solve/clarify several issues as detailed below.

  1. Senescence cancer cells should be briefly discussed, as senescence also leads to non-proliferation of cancer cells. Authors should explained how to differentiate between quiescent and senescent cancer cells, as many of the methods the authors are describing will not differentiate between the two states.
  2. Authors should explain that cancer cells are on average less proliferative in vivo than in a typical 2D in vitro cell culture where the growth is usually exponential due to availability of growth factors and nutrients in culture media.
  3. For in vitro experiments, the authors should explain how to differentiate QCC from normal quiescent cells in tissues.
  4. Is there any identification of QCC in the publicly available scRNAseq datasets from human tumor samples/animal experiments? The authors should discuss this aspect, as scRNAseq could potentially be quite useful in identifying QCC cells in vivo.
  5. Authors should emphasize that even their ‘homogenous’ models do not contain only QCC. Perhaps it would be more exact to say that QCC are enriched in these model?
  6. As a distinction is made between homogenous and heterogeneous models of QCC, these two groups should be organize under separate subheadings. Now both homo- and heterogenous models are lumped together.
  7. Membrane labeling dyes are just dyes that go in the membrane and work on a similar principle as other dyes, there is dilution of signal with cell divisions. They should not have a separate heading, and be included in the section ‘3.5. Pulse-chase identification’ instead.
  8. The DNA replication indicators BrdU and EdU should moved to the DNA/RNA section, as they are closely related, and specifically concern DNA. The combination of BrdU/EdU labeling and DNA content is frequently used to asses quiescence, the execution is similar to the Pyronin Y/Hoechst method.
  9. Line 336: Please note that acridine orange also stains lysosomes. This information should be included in order not to lead the reader astray. Pyronin Y/Hoechst is likely a superior method to acridin eorange.
  10. Line 420: Hypoxic markers such as pimonidazol and Hif1a mark hypoxia. QCC may be in hypoxic areas, true, but they are not specifically marked by these stainings.

Author Response

Response to Reviewer 2 Comments

Point 1: Senescence cancer cells should be briefly discussed, as senescence also leads to non-proliferation of cancer cells. Authors should explained how to differentiate between quiescent and senescent cancer cells, as many of the methods the authors are describing will not differentiate between the two states. 

Response 1: We agree with the reviewer’s assessment. We have described the difference between quiescent and senescent cancer cells in the “1.0 Introduction” (on page 1, line 41) and methods to differentiate between them in “3.1 Ki-67” section (on page 11, line 389).

Point 2: Authors should explain that cancer cells are on average less proliferative in vivo than in a typical 2D in vitro cell culture where the growth is usually exponential due to availability of growth factors and nutrients in culture media.

Response 2: Yes, we agree. On page 9 (line 306), we have added about the growth difference between 2D culture and in vivo in the section “2.3 In vivo quiescent models”.

Point 3: For in vivo experiments, the authors should explain how to differentiate QCC from normal quiescent cells in tissues.

Response 3: Agreed. On page 9 (line 300), we have included about methods to differentiate between QCC and normal quiescent cells in tissue in “2.3 in vivo quiescent models” section.

Point 4: Is there any identification of QCC in the publicly available scRNAseq datasets from human tumor samples/animal experiments? The authors should discuss this aspect, as scRNAseq could potentially be quite useful in identifying QCC cells in vivo.

Response 4: The suggestion is interesting. Concordantly, we have searched for publicly available scRNA-seq dataset and found none. However, we found several articles that presented scRNA-seq findings for QCCs in human tumor sample and animal experiment. We therefore have included this finding in “3.6 Discussion” section (page 14, line 535) and describe the potential of scRNAseq.

Point 5: Authors should emphasize that even their ‘homogenous’ models do not contain only QCC. Perhaps it would be more exact to say that QCC are enriched in these model?

Response 5: Agreed. On page 2 (line 84), we have rephrased the sentence in “2. Model mimicking QCC” to state that QCCs are enriched in the homogenous model.

Point 6: As a distinction is made between homogenous and heterogeneous models of QCC, these two groups should be organize under separate subheadings. Now both homo- and heterogenous models are lumped together.

Response 6: Agreed. We have separated the models into the “homogenous” and “heterogenous models of QCC”.

Point 7: Membrane labeling dyes are just dyes that go in the membrane and work on a similar principle as other dyes, there is dilution of signal with cell divisions. They should not have a separate heading, and be included in the section ‘3.5. Pulse-chase identification’ instead.

Response 7: Agreed. We have incorporated the membrane labelling dyes into the “3.5 Pulse chase identification” section (page 12, line 469).

Point 8: The DNA replication indicators BrdU and EdU should moved to the DNA/RNA section, as they are closely related, and specifically concern DNA. The combination of BrdU/EdU labeling and DNA content is frequently used to asses quiescence, the execution is similar to the Pyronin Y/Hoechst method.

Response 8: Agreed. We have moved the description on BrdU and Edu into the “3.2 RNA and DNA content” section (page 11, 409).

Point 9: Line 336: Please note that acridine orange also stains lysosomes. This information should be included in order not to lead the reader astray. Pyronin Y/Hoechst is likely a superior method to acridine orange.

Response 9: Agreed. On page 11 (line 407), we have added information about acridine orange stain lysosomes in the “3.2 RNA and DNA content” section.

Point 10: Line 420: Hypoxic markers such as pimonidazol and Hif1a mark hypoxia. QCC may be in hypoxic areas, true, but they are not specifically marked by these staining.

Response 10: Agreed. On page 12 (line 444), we have added information to clear the non-specificity of hypoxic markers in the “3.4 Potential QCC molecular markers” section.

Reviewer 3 Report

Re: “Towards a framework for better understanding of quiescent 2 cancer cells” - review

This submitted manuscript describes an important field of cancer biology, dormancy phenomena. Even though the authors have trying to update the existing models and the detection methods for quiescent cancer cells (QCCs), the manuscript needs considerable further work to be accepted for publication.

Major concerns:

The first impression I have reading this manuscript is the lack of integration between the main concept and what can be done with the experimental models and the ways to track QCCs.

Overall, section 2, “Models mimicking quiescent cancer cells”, is presented as an enumeration without any discussion of how to decide to use a model, or what method should be choose  for detection based on a certain model/hypothesis. There is no discussion on the properties of cell lines described in table 1 which make them potentially harboring QCCs.

Section 2.5, “Anticancer-treatment-induced quiescence, misses to discuss a very important concept in cancer biology. Does anticancer treatment induce QCCs (as authors claim) and/or anticancer treatment by killing majority of cancer cell induce a clonal selection of non-proliferating cells (aka, QCCs).

Cancer cell lines are not the best model to describe and study QCCs. These immortalized adapted cells do not reflect the real behavior of QCCs in animals with spontaneous cancer (not syngeneic models developed upon external inoculation of a murine cell line) or, more important, in humans. The natural history of an “exogenous” cancer is far by the real dynamic of different sub-compartments of cancer cells in a host (cancer derived from the host). Moreover, I do not have any clue why there is not information in this manuscript about the impact of immunity on cancer quiescence.

Immune incompetent mice are the worst case-scenario to study QCCs since miss a major host compartment which interfere with QCCs natural history, the immune system.

I believe that for avoiding any confusion in reader mind, a short discussion about the differences and similarities between quiescent cancer stem cells and QSCs should be added.

Finally, there are many interesting models of dormancy which are not presented in this work (e,g, Uhr J and Vitetta E’s humngous work in cancer dormany).

Author Response

Response to Reviewer 3 Comments

Point 1: Lack of integration between the main concept and what can be done with the experimental models and the ways to track QCCs.

Response 1: We agreed with your constructive comments. We therefore have added the detection method in Table 1 to integrate between the experimental model and detection method of QCCs. On page 14 (line 556), we have also added some discussion in “4. Conclusion and future perspectives” section, to better integrate between them.

Point 2: Overall, section 2, “Models mimicking quiescent cancer cells”, is presented as an enumeration without any discussion of how to decide to use a model, or what method should be choose for detection based on a certain model/hypothesis.

Response 2: On page 10 (line 332), we have added about how to select QCC model in the “2.4 Discussion” section. We have also added the detection method used in each model in Table 1.

Point 3: There is no discussion on the properties of cell lines described in table 1, which make them potentially harboring QCCs.

Response 3: Quiescence is a cellular state and all cells (regardless normal or malignant) have the potential to adopt the quiescent or G0 state. In an adult, most cells are in a G0 state. In cancer, having cells at G0 is a norm not an exception, although there appears to have a variation in the proportion of G0 cells. The mechanism underlying the variation is not clear.

Point 4: Section 2.5, “Anticancer-treatment-induced quiescence, misses to discuss a very important concept in cancer biology. Does anticancer treatment induce QCCs (as authors claim) and/or anticancer treatment by killing majority of cancer cell induce a clonal selection of non-proliferating cells (aka, QCCs).

Response 4: Agreed. On page 6 (line 158), we have clarified about it in “2.1.4 Anticancer treatment increase proportion of QCC” section.

Point 5: Cancer cell lines are not the best model to describe and study QCCs. These immortalized adapted cells do not reflect the real behavior of QCCs in animals with spontaneous cancer (not syngeneic models developed upon external inoculation of a murine cell line) or, more important, in humans. The natural history of an “exogenous” cancer is far by the real dynamic of different sub-compartments of cancer cells in a host (cancer derived from the host).

Response 5: Agreed. On page 10 (line 346), we have clarified about it in the “2.4 Discussion” section.

Point 6: Moreover, I do not have any clue why there is not information in this manuscript about the impact of immunity on cancer quiescence.

Response 6: Immunity is an important aspect of cancer quiescence. However, as the relevant literature is relatively limited at present and this review mainly focus on the model and detection method for quiescent cancer cells, we therefore did not include it in our review.

Point 7: Immune incompetent mice are the worst case-scenario to study QCCs since miss a major host compartment, which interfere with QCCs natural history, the immune system.

Response 7: Yes, we agree with the reviewer. Kindly refer our response at No. 6 above.

Point 8: I believe that for avoiding any confusion in reader mind, a short discussion about the differences and similarities between quiescent cancer stem cells and QCCs should be added.

Response 8: Agreed. On page 2 (line 58), we have clarified about quiescent cancer stem cells and QCCs in the “1.0 Introduction” section.

Point 9: Many interesting models of dormancy that are not presented in this work (e.g. Uhr J and Vitetta E’s humongous work in cancer dormancy.

Response 9: On page 9 (line 282), we have added the researcher’s work in “2.3 In vivo quiescent models” reference 49 and 57.

Reviewer 4 Report

Wan Najbah Nik Nabil et al provide a review on the various techniques that allow modeling of QCC. Moreover, they include descriptions of the most commonly used methods to determine whether cells are indeed quiescent. Models and methods are well explained and easy to understand, however, in order to make the review easier to understand by non-experts and provide a better overview to those working in dormancy, a few aspects need improvement:

  • Before explaining all the models for QCC, a short overview of markers and characteristics of QCC in patients should be given. More context is necessary to help readers understand how each model can help in the study of these cells. Are QCC only found as disseminated cells? Or can they be found in tumor masses? This is important to discuss because disseminated cells in solitude are very different from tumor cells in a petri dish.
  • For each model, advantages and disadvantages should be mentioned and more importantly, what specific aspect of the phenomenon the technique is modeling. For example, how is nutrient deprivation modeling disseminated dormant tumor cells in a patient? What specific aspect of the process or phenomenon?
  • For each method to detect QCC, it needs to be explained at the beginning whether that method is for detection on G0 cells in a petri dish, in a mouse model on in a tumor in patients. For example, Ki67, is that on tissue sections my IHC or IF? Flow cytometry? Or is it quantified by flow cytometry (since authors suggest combination with PI). This can be very confusing by readers searching for methodologies to use in their work and all these differences must be explained.
  • In in vivo models, work from Aguirre-Ghiso and Aznar-Benitah should be included.

Author Response

Response to Reviewer 4 Comments

Point 1: Before explaining all the models for QCC, a short overview of markers and characteristics of QCC in patients should be given.

Response 1: Agreed. On page 1-2, we have added in the “1.0 Introduction” section about the characteristic of QCCs in patients (such as being in G0 phase, reversibility from quiescence to proliferation, surviving harsh environment) and an overview of the QCC markers.

Point 2: More context is necessary to help readers understand how each model can help in the study of these cells. Are QCC only found as disseminated cells? Or can they be found in tumor masses? This is important to discuss because disseminated cells in solitude are very different from tumor cells in a petri dish.

Response 2: Agreed. On page 2 (line 49), we have added in the “1. Introduction” about the development of cancer cells and try to relate with the experimental models. We have also added that QCCs can be found in tumor masses and disseminated cell (page 2, line 61).

Point 3: For each model, advantages and disadvantages should be mentioned.

Response 3: Agreed. The application and limitation of each models have been described in Table 1.

Point 4: And more importantly, what specific aspect of the phenomenon the technique is modeling. For example, how is nutrient deprivation modeling disseminated dormant tumor cells in a patient? What specific aspect of the process or phenomenon?

Response 4: We agree that overall, there is a gap of experimental modelling QCC phenomenon. Regarding nutrient deprivation model, on page 2 (line 55), we have added that QCC in vitro experimental model of nutrient deprivation and hypoxia-induced mimic the certain clinical aspects of QCC, namely nutrient starvation and oxygen deprivation of QCC that situated far away from blood vessels.

Point 5: For each method to detect QCC, it needs to be explained at the beginning whether that method is for detection on G0 cells in a petri dish, in a mouse model on in a tumor in patients. For example, Ki67, is that on tissue sections my IHC or IF? Flow cytometry? Or is it quantified by flow cytometry (since authors suggest combination with PI). This can be very confusing by readers searching for methodologies to use in their work and all these differences must be explained.

Response 5: Agreed. For clearer understanding whether the QCCs detection method are for in vitro or in vivo, we have added the detection method used in the corresponding QCCs model, in Table 1 (page 3).

Point 6: In in vivo models, work from Aguirre-Ghiso and Aznar-Benitah should be included.

Response 6: Thank you for suggesting above researchers. On page 8, we accordingly have included reference 46, the work of Aguirre-Ghiso in Table 1 and “2.3 In vivo quiescent models” section (page 8, line 262). However, we did not include the work of Aznar-Benitah as the researcher mainly studying about stem cell, which is a type of cells with its unique transcription factors, selective location in a tissue and may not be exclusively residing at Go phase.

Round 2

Reviewer 1 Report

The authors responded correctly to requests for improvement.

Reviewer 2 Report

The authors implemented changes suggested by the reviewers. A minor note: The newly-introduced sections would benefit from some additional language corrections.  

Reviewer 3 Report

I would like to thank the authors for their outstanding work to answer my questions. The revised manuscript is responsive to my previous comments including adding new important information and comprehensive integration of these new data. I believe that this updated manuscript is suitable for publication now.

Reviewer 4 Report

the review has improved in clarity to understand how each experimental approach models the phenomenon of QCC in patients. It also clarifies the distinction between quiescence and senescence, which is essential and often confuses those outside the immediate field.